# Comparing Conventional Physician-Led Education with VR Education for Pacemaker Implantation: A Randomized Study

**DOI:** 10.3390/healthcare12100976

**Published:** 2024-05-09

**Authors:** Adela Drozdova, Karin Polokova, Otakar Jiravsky, Bogna Jiravska Godula, Jan Chovancik, Ivan Ranic, Filip Jiravsky, Jan Hecko, Libor Sknouril

**Affiliations:** 1Department of Cardiology, Agel Hospital Trinec-Podlesi, 739 61 Trinec, Czech Republic; 2Faculty of Medicine, Masaryk University, 601 77 Brno, Czech Republic; 3Faculty of Medicine, Palacky University, 779 00 Olomouc, Czech Republic; 4Faculty of Medicine, University of Ostrava, 701 03 Ostrava, Czech Republic; 5Philosophical Faculty, Masaryk University, 601 77 Brno, Czech Republic; 6Faculty of Electrical Engineering and Computer Science, VSB-Technical University of Ostrava, 708 00 Ostrava, Czech Republic

**Keywords:** patient education, virtual reality, pacemaker implantation, medical technology in education, health outcomes, preoperative anxiety

## Abstract

Introduction: Education of patients prior to an invasive procedure is pivotal for good cooperation and knowledge retention. Virtual reality (VR) is a fast-developing technology that helps educate both medical professionals and patients. Objective: To prove non-inferiority of VR education compared to conventional education in patients prior to the implantation of a permanent pacemaker (PPM). Methods: 150 participants scheduled for an elective implantation of a PPM were enrolled in this prospective study and randomized into two groups: the VR group (*n* = 75) watched a 360° video about the procedure using the VR headset Oculus Meta Quest 2, while the conventional group (*n* = 75) was educated by a physician. Both groups filled out a questionnaire to assess the quality of education pre- and in-hospital, their knowledge of the procedure, and their subjective satisfaction. Results: There was no significant difference in the quality of education. There was a non-significant trend towards higher educational scores in the VR group. The subgroup with worse scores was older than the groups with higher scores (82 vs. 76 years, *p* = 0.025). Anxiety was reduced in 92% of participants. Conclusion: VR proved to be non-inferior to conventional education. It helped to reduce anxiety and showed no adverse effects.

## 1. Introduction

Patient education is a key factor in ensuring good cooperation and adherence to a recommended treatment plan. A well-educated patient is more likely to be satisfied with their care and will be more compliant [1]. However, it may be difficult for them to retain the message if the information is too sophisticated or elaborate. This was highlighted in a review proving that around half of patients (40–60%) cannot correctly recall the information given by their physician within 10–80 min from their session [1]. There was also a high percentage of misunderstanding (up to 60%) of the information provided directly after the session [2].

Education plays an important role in reducing patients’ anxiety about their condition and/or the planned intervention. The fear of unfamiliar environments, staff, and procedures may be debilitating. Anxiety related to cardiac procedures is well described in the medical literature as an excessive fear with a sense of imminent jeopardy, and can occur before and/or during the procedure [2]. Studies suggest that up to half of patients undergoing percutaneous coronary intervention suffer from significant anxiety [2]. A lack of knowledge of the procedure and insufficient anesthesia are the main reasons for periprocedural anxiety [2]. Anxiety causes cardiac autonomic dysfunction, leading to raised blood pressure and heart rate and reduced heart rate variability [3]. It can also mimic cardiac symptoms like chest tightness, breathlessness, and palpitations, which can mislead the operator during the procedure. Anxiety was found to be an independent predictor for postoperative cardiovascular outcomes and 4-year mortality after cardiac procedures [4]. It is also linked to higher cardiovascular morbidity and mortality after cardiac procedures [4].

Virtual reality (VR) offers several advantages for patients’ education: it immerses patients directly in a real-time simulation, it is interactive, and it is entertaining; all of this contributes to enhanced information retention. This innovative technology can be used in rehabilitation as part of an exercise program [5], and in pediatric care to reduce anxiety by providing information about the procedure or by serving as a distraction [6]. Virtual reality is well established in the field of educating health care professionals [7]. It is also an emerging tool for the education of patients, either prior to invasive procedures or as part of a treatment plan for specific conditions [8,9,10,11,12].

Within cardiology, reports were given on the use of VR prior to a percutaneous coronary angiography [13], atrial septal closure [14], or ablation of atrial fibrillation [15]. To our knowledge, no study was performed to educate patients prior to an implantation of a permanent pacemaker (PPM). This procedure is predominantly performed as a day case in the non-acute setting; patients are frequently insufficiently educated by their primary care physicians and undergoing the procedure requires certain post-operative precautions. The PPM implantation is often performed only with the use of local anesthetics; sedation can be used as well. It is therefore crucial for the patient to understand the procedure well to remain calm during the surgery and to reduce their anxiety levels for the best outcomes. The population requiring this procedure is generally older and there are limited data on use of VR in an older population [16,17]. There are reports where VR was used to help with mobility and balance [18] or with mood [19].

We aim to prove the feasibility and non-inferiority of patients’ education with the use of VR prior to the implantation of a PPM and its beneficial effects on anxiety.

## 2. Materials and Methods

### 2.1. Study Design and Participants

This study was registered with ClinicalTrials.gov (NCT05695534). It is a prospective, randomized control study with two parallel groups. In total, 150 patients were included in the study. This includes all the patients who underwent an elective permanent pacemaker (PPM) implantation as a one-day procedure via our cardiac Suite from January to December 2023. Exclusion criteria were significant visual impairment or significant cognitive impairment that would prevent the patient from filling out the questionnaire. 

### 2.2. Randomization and Blinding

The participants were randomized into two groups: One group received the education about the procedure provided by the hospital staff and signed a standard informed consent form (a conventional group). The other group watched a video in virtual reality, instead of receiving the standard education, and then signed the consent form (a VR group). The randomization was done using an envelope method, with blocks of 30 patients. Sealed opaque envelopes were randomly selected by cardiac Suite staff. The hospital staff and the participants were not blinded to the results of randomization. 

### 2.3. Interventions

A team of physicians created a screenplay containing all the necessary information about the procedure. The 360° video was recorded at our hospital premises, with our staff cast, using camera Insta X2 (Shenzen, China) in 5,6K resolution with 30 FPS. The movie was finalized in Adobe Premiere and presented in standalone wireless VR headset Oculus Meta Quest 2 (Reality Labs, Meta Platforms, MenloPark, CA, USA). The six-minute-long movie presents the hospital with its admission suite, cardiac suite, and electrophysiology theatre that features equipment designed to reduce anxiety in patients unfamiliar with the environment. The procedure, including its complications, is explained with the use of animations and pictures in the background to enhance understanding. The conventional and VR educations of participants were both performed in the cardiac suite in a quiet and safe environment. Both groups of patients had the opportunity to ask questions whenever they needed for better understanding. All the educational materials were provided in the Czech language to ensure the best understanding of the participants. To ensure the highest video quality and the most accurate educational content, a group of physicians involved in the procedure reviewed the VR content. We tested several different headsets to find the best combination of quality, comfort for the participants, battery life, software stability, and user friendliness for the operating staff. 

Both groups of patients had the opportunity to ask questions after receiving the education. They subsequently filled out a questionnaire composed of questions about their pre-hospital education, their understanding after our education, and their experience with the VR. 

### 2.4. Outcomes

The questionnaire assessed the quality of both pre-hospital and in-hospital education using a five-point Likert scale, where 1 indicated “very satisfied” and 5 indicated “very dissatisfied.” The scale was designed to mirror the grading system used in the Czech education system, making it more intuitive for the participants.

The questionnaire also included four multiple-choice questions to evaluate the participants’ understanding of the procedure. These questions focused on key aspects of the pacemaker implantation process, such as:Type of anesthesia used during the procedure (local anesthesia, general anesthesia, or sedation)Location of the pacemaker pocket (left or right side of the chest)Potential complications associated with the procedure (infection, bleeding, lead dislocation, or pneumothorax)Whom to contact in case of complications after the procedure (general practitioner, cardiologist, or hospital emergency department)

An Educational Impact Score was calculated based on the number of correct answers to these four questions, with a maximum possible score of 4 points.

Additionally, the subjective experience of the VR video was examined using two closed-ended questions:“Did the video help you to reduce your anxiety levels before the procedure?” (Yes/No)“Was watching the VR video a positive experience?” (Yes/No)

These questions aimed to assess the potential benefits of VR education in reducing pre-procedural anxiety and improving patient satisfaction.

### 2.5. Adverse Effects

Participants were educated about potential adverse effects of the VR (dizziness/vertigo, disorientation, and falls) and signed the consent form. They were seated in a quiet area to watch the video and were advised not to stand up either during the educational video or shortly after its finish to prevent falls.

### 2.6. Statistical Analysis

Statistical analysis was performed using IBM SPSS Statistics for Windows, Version 29.0 (Armonk, NY, USA: IBM Corp.). The primary outcome was the quality of education, with secondary outcomes including knowledge retention, anxiety reduction, and patient satisfaction. These outcomes were pre-specified in the study protocol. Continuous variables were analyzed using the Mann–Whitney U test, which was chosen for its non-parametric nature suitable for our data distribution. Categorical variables were assessed using Chi-square tests to determine the association between them. Fisher’s exact test was utilized for educational-impact scores, where small sample sizes may affect the accuracy of the Chi-square test.

Baseline characteristics of the study population were presented as medians with interquartile ranges (IQR) for continuous variables and as counts (N) and percentages (%) for categorical variables. The quality of pre-hospital and in-hospital education was evaluated using a five-point Likert scale, with 1 indicating the highest quality and 5 the lowest. Knowledge retention was assessed using four multiple-choice questions, and an Educational Impact Score (ranging from 0 to 4) was calculated based on the number of correct answers.

To further analyze the factors influencing educational outcomes, patients were divided into two subgroups based on their Educational Impact Scores: a limited-comprehension group (scores 1–2) and a higher-impact group (scores 3–4). Differences in age, gender, and education type between these subgroups were investigated using the Mann–Whitney U test for age and Chi-square tests for gender and education type. All statistical tests were two-sided, and a *p*-value < 0.05 was considered statistically significant. 

## 3. Results

### 3.1. Baseline Characteristics

One hundred and fifty participants were enrolled into the study; all of them were Caucasian. They were randomly selected to receive the conventional education (*n* = 75) or the VR education (*n* = 75). All the enrolled patients completed the education prior to their procedure. None of the patients from the VR group had previous experience with this technology. The baseline characteristics of our cohort (age and sex) were balanced after randomization and can be found in the Table 1. The median age was 76 (IQR 70–83), with no significant age difference between the two subgroups. The cohort was 57.3% male. The VR group and the conventional group consisted of 41 men and 34 women and 45 men and 30 women, respectively.

### 3.2. Quality of Pre-Hospital Education

Formal education by a referring physician prior to an admission was satisfactory in both groups; most patients reported either Mark 1, as the best score (63.3%), or Mark 2 (31.3%). There was only one patient who reported Mark 5, as the worst score or if no education was given. There were no statistical differences between the two groups, as per the Table 1.

### 3.3. Quality of In-Hospital Education

A majority of patients reported better understanding after being educated in the hospital (97.3%), with no significant difference between the two groups (96% vs. 98.7% in the VR and conventional group, respectively). The quality of hospital education was evaluated; 85.3% of patients rated it as the best, Mark 1, and the rest of the results can be found in Table 2. From the VR group, 86.7% of participants reported the quality of education (VR video) as the best. In the conventional group, 84% of patients rated the education as the best. There was no significant difference in the evaluation of education between the two groups. 

### 3.4. Subjective Experience

Ninety two percent of patients (*n* = 69) reported reduced levels of anxiety after watching the VR video and seeing where the procedure was going to take a place. The rest of the patients stated they were not anxious in the first place. All the patients reported that watching the VR video was a positive experience. None of the patients complained of vertigo or other intolerance of the VR, and no adverse effects were reported. Four patients mentioned they had problems hearing some parts of the audio content.

### 3.5. Assessment of Knowledge

Successful answers to the knowledge-based questions were examined by calculating the Educational Impact Score, as shown in Table 3. There was no significant difference between the two groups of patients. Sixty-five patients had all four questions answered correctly (43.3%), with 33 (44%) and 32 (42.7%) patients in the VR group and the conventional group, respectively. Three correct answers were given by 36 patients in the VR group (48%) and by 33 patients in the conventional group (44%). A score of zero was achieved by only one patient in the conventional group. 

A subsequent analysis was made by dividing the patients into two subgroups based on their achieved score: a limited-comprehension group (score 1–2) and a higher-impact group (score 3–4). There was a significant difference in age between these groups, with a median of 82 years of age in the limited-comprehension group, and a median of 76 years of age in the higher-impact group (see Table 4). No significant difference was found in the distribution of gender or the type of education received (conventional vs. VR enhanced). Out of the four questions, patients most frequently failed to answer the question about whom to contact in case of complications; only 48.7% of patients answered correctly (52% in the VR group and 45.3% in the conventional). With the other answers, the results were more balanced, as shown in Table 5. There appears to be a trend towards higher percentages of correct answers in the VR-enhanced group; however, the results did not reach statistical significance. 

## 4. Discussion

Patient education plays a key role in peri- and post-procedural cooperation, adherence to recommendations, recovery, and overall experience of the procedure [20,21]. Pre-procedural education was linked to better managing the actual procedure [20,21]. Printed brochures with pictures and diagrams, educational videos, and information booklets have been used over the last years to facilitate understanding and help with education [20]. The retention of information by patients is, however, unsatisfactory [1]. Virtual reality is proving to be a good way to increase patients’ interest, engagement, and retention of information [11].

Our study aimed to assess both subjective experience and knowledge retention in patients undergoing implantation of a permanent pacemaker as an elective day case. These patients are typically older, as conduction system disorders requiring pacing are more frequent at higher ages [22]. Our population, therefore, is much older compared to other VR studies, where the mean age ranged from 40 to 65 years [11]. Despite this, no adverse effects were reported within our participants. Patients generally reported a positive experience of the VR, consistent with the results of Huygelier et al. [17], who assessed the attitudes of older populations towards novelty technologies. Unfortunately, there are no studies focusing on educating an older population with the use of VR. The only available studies focus on mood improvement [19] or improvement of patients’ mobility and balance [18]. A majority of patients from our cohort stated that VR helped them to understand the procedure better. The scores for perceived quality of education did not differ between the VR and the conventional group. As stated before, VR should enhance knowledge retention [11], and better results should be, therefore, expected in the VR group when assessing the results of the knowledge-based questions. This was confirmed in studies using VR education before cancer treatment [23], cardiothoracic surgery [24], or atrial fibrillation ablation [15]. This was, however, not the case of our study; additionally, although there was a trend towards a higher educational score in the VR group, the results were not statistically significantly different. It is possible that, with a higher number of participants, this difference would reach statistical significance. One explanation for failure in reaching significant difference could be the fact that the older population, in general, can have problems retaining given information and understanding [25]. This theory is supported by the fact that patients with lower scores (the limited-comprehension group) were generally older than patient with higher scores, the difference was statistically significant. The other explanation could be that the questionnaire, with multiple choice answers, was not comprehensible enough for the older population. Impaired hearing might be another explanation why patients’ comprehension was not better, as four patients in our cohort reported they had difficulties hearing the audio properly. Gender did not significantly influence the results and no further reports were found where the influence of gender on VR education was studied.

As the results suggest, there was one question that was problematic: Whom should you contact in case of complications? Even though the importance of contact information is highlighted in our educational materials, and was specifically emphasized in the VR video, almost half of our participants answered incorrectly. The fact that this information is given at the very end of the video, where attention of participants could be fading, might be one of the explanations. This analysis provides valuable feedback for our practice, suggesting we need to stress this information even more. 

It is reassuring that a majority of patients received good education by the referring physician, as we sometimes experience patients who claim to have received no or very little education in the pre-hospital setting. The VR group’s perception of in-hospital education quality was not statistically different from that of the conventional group, supporting the non-inferiority of VR education. 

VR has been used to reduce anxiety for a long time, especially in pediatric patients [6]. We believed that even adults experience anxiety when coming into a new hospital for an unknown procedure. We therefore filmed our 360° video in the hospital setting (admission suite, cardiac suite, and theatres) to enable patients to get to know the hospital better prior to the procedure. Real doctors and nurses star in the video to make the experience as authentic as possible. This is to help patients overcome their fear of unknown. By explaining the procedure in detail, with the use of animations and pictures, we target the fear of the procedure that stems from a lack of knowledge. As anxiety might be one of the factors contributing to poorer outcomes, and one of the factors complicating the surgery for the operator [2,4,26], we tried to reduce it with the VR video. As 92% of participants reported reduced anxiety after watching the video (with the rest stating they were not anxious), the results suggest that this might be a promising modality for future. 

Finally, a physician-led education is time consuming and, therefore, expensive. VR may offer an alternative way to educate patients not only prior to procedures, but also about different treatment plans [8,10,12]. VR education has the potential to save significant costs and free up valuable physician time. Further studies designed to assess the non-inferiority of VR education in larger groups of patients are, however, necessary to enable widespread use of this modality.

Based on our study’s results and the experience gained during the implementation of VR-based patient education, we offer the following recommendations for healthcare providers considering the use of VR in educating older adults:(a)Duration of VR sessions: Our study used a 6-min video, which was well-tolerated by the participants. We recommend keeping VR sessions concise, preferably under 10 min, to maintain patient comfort and minimize the risk of adverse effects such as disorientation or fatigue.(b)Age-appropriate content and delivery: When creating VR content for older adults, it is crucial to use clear, easily understandable language and age-appropriate visual elements. We found that a combination of narration, simple text, and engaging graphics helped in effectively conveying the educational message. Moreover, we suggest using a 360° video format, as it provides an immersive experience without the need for complex interaction, which may be more suitable for older individuals.(c)Minimizing adverse effects: To prevent potential side effects, such as motion sickness or disorientation, we advise ensuring that patients are seated comfortably and securely during the VR session. Continuously monitor patients for any signs of discomfort and be prepared to pause or terminate the session if necessary. Designing the VR environment with minimal rapid movements and transitions can also help reduce the risk of adverse effects.(d)Ongoing patient engagement and feedback: Regularly seek feedback from older patients regarding their VR education experience. Encourage them to share any difficulties, concerns, or suggestions for improvement. Use this valuable input to refine and optimize the VR content and delivery methods iteratively, ensuring that the educational intervention remains tailored to the specific needs of older adults.(e)Future studies should explore the adaptation of VR content for various languages and cultural contexts to increase the generalizability of the findings. In our study, the VR content was tailored to our hospital setting, featuring our staff, theatres, and cardiac suite, which may have helped patients feel more familiar and comfortable. However, we acknowledge that our participant population was predominantly Czech-speaking, and the content was not designed to cater to a diversity of linguistic or cultural backgrounds. Adapting the VR content to include multiple language options and culturally relevant elements could broaden its applicability and effectiveness in diverse patient populations.

This single-center study has several limitations. One of them is a small sample size, as many patients requiring PPM implantation come in as acute cases and, therefore, cannot be included in the study. This can increase the risk of a Type 2 error. We have not studied the effect of participants’ education on the results; this could have an impact on patients’ understanding of the educational information and the questions, and knowledge retention might also be better with prolonged education [25]. Only immediate follow-up was performed as a part of this study, and the lack of further long-term follow-up to assess potential complications is another limitation. The study could not be blinded due to the character of the intervention. With the questionnaire assessing subjective experience and comprising unvalidated knowledge-based questions, this study could be subject to inherent bias. Due to the tertiary nature of our center, with patients coming from a broad area, it would be challenging to conduct a longitudinal follow-up. 

VR education proves to be feasible even in the older population, but attention needs to be given to ensuring adequate audio so that even patients with hearing loss can understand. We had a good experience with the 360° video format, which seemed to cause none of the significant side effects that watching 3D videos can sometimes have. However, even in the setting of virtual reality, important information needs to be emphasized to ensure its retention. VR appears to be a valuable tool to reduce procedure-related anxiety. At the current level, the widespread use might be challenging; however, as the technology evolves very quickly, VR education might replace standard education in the coming years. But, finally, while we believe that VR education may have limited direct influence on peri-procedural complications, it could potentially impact wound healing and lead dislocation.

## 5. Conclusions

Education, prior to the implantation of a PPM, with the use of VR was shown to be non-inferior to conventional education led by a physician. It is feasible even in an older population without frequent adverse effects. VR education relieves patients’ anxiety and improves their understanding of the procedure. There was a trend towards higher educational scores in the VR education group. The results suggest that age was the major determinant of knowledge retention in the older population undergoing implantation of a PPM. Regardless of the type of education, older patients achieved lower scores when their knowledge was assessed after education. Important information needs to be stressed to ensure its remembering, even in the setting of virtual reality. 

## Figures and Tables

**Table 1 healthcare-12-00976-t001:** Baseline Characteristics of Patients Randomized in Conventional vs. VR Education.

		Total		VR Education		Conventional Education		*p*-Value
Total (N)		150		75		75		
Age (median; [IQR])		76	[70–83]	77	[68–84]	76	[70–81]	0.578
Sex (N; %)	Male	86	57.3%	41	54.7%	45	60.0%	0.621
	Female	64	42.7%	34	45.3%	30	40.0%	
Referral Educational Quality (N; %)	1	95	63.3%	47	62.7%	48	64.0%	
	2	47	31.3%	23	30.7%	24	32.0%	
	3	6	4.0%	3	4.0%	3	4.0%	0.73
	4	1	0.7%	1	1.3%	0		
	5	1	0.7%	1	1.3%	0		

Note: Age is presented as the median with the interquartile range [IQR] in brackets. N represents the number of participants and % indicates the percentage of the total group. Referral Educational Quality is assessed on a Likert scale from 1 (highest quality) to 5 (lowest quality). *p*-values are derived from Chi-square tests for categorical variables and the Mann–Whitney U test for continuous variables, where a *p*-value < 0.05 indicates statistical significance.

**Table 2 healthcare-12-00976-t002:** Physician Education Quality and Efficacy of Physician-Led Education in Improving Patient Understanding: Total Cohort and Study Arm Comparison.

		Total		VR Education		Conventional Education		*p*-Value
Education Quality Score (N; %)	1	128	85.3%	65	86.7%	63	84.0%	0.243
	2	20	13.3%	8	10.7%	12	16.0%	
	3	2	1.3%	2	2.7%	0		
	4 or 5	0		0		0		
In-Hospital Education Added Value (N; %)	Not Improved	4	2.7%	3	4.0%	1	1.3%	0.311
	Improved	146	97.3%	72	96.0%	74	98.7%	

Note: Education Quality Score is based on a Likert scale from 1 (best) to 5 (worst) to assess the perceived quality of education received. ‘In-Hospital Education Added Value’ reflects participants’ perceptions of improvement after Study-Arm Education compared to their baseline. *p*-values are calculated using Chi-square tests for categorical variables to determine the statistical significance of differences between VR and conventional education groups. A *p*-value < 0.05 would indicate a statistically significant difference.

**Table 3 healthcare-12-00976-t003:** Comparison of Educational Impact Scores Between VR and Conventional Education Groups.

	Score	Total		VR Enhanced		Conventional		*p*-Value
Total			150	75		75		
Educational Impact Score (N; %)	4	65	43.3%	33	44.0%	32	42.7%	0.873
	3	69	46.0%	36	48.0%	33	44.0%	
	2	10	6.7%	4	5.3%	6	8.0%	
	1	5	3.3%	2	2.7%	3	4.0%	
	0	1	0.7%	0	0.0%	1	1.3%	

Note: The table delineates the distribution of Educational Impact Scores, which range from 0 (no impact) to 4 (highest impact), for patients in both the VR and conventional education groups following the pacemaker procedure study. The counts (N) and percentages (%) illustrate the proportion of patients achieving each score level. The *p*-value reported is based on Fisher’s exact test, which is utilized here due to the small sample size in some score categories. A *p*-value of 0.873 indicates there is no statistically significant difference between the two groups in terms of educational impact at the 0.05 significance level.

**Table 4 healthcare-12-00976-t004:** Characteristics Comparison by Educational Impact of Pacemaker Education.

		Total		Limited Comprehension		Higher Impact		*p*-Value
Age (median; [IQR])		76	[70–83]	82	[76–87]	76	[69–82]	0.025
Sex (N; %)	Male	86	57.3%	7	43.8%	79	59.0%	0.245
	Female	64	42.7%	9	56.3%	55	41.0%	
Education Type (N; %)	VR Enhanced	75	50.0%	6	37.5%	69	51.5%	0.29
	Conventional	75	50.0%	10	62.5%	65	48.5%	

Note: This table contrasts the baseline characteristics of patients grouped by their level of educational outcome after pacemaker education: those with limited comprehension (scores of 0–2) and those with a higher impact (scores of 3–4). The median age and interquartile range (IQR) are reported. A significant difference in median age was observed between the two groups (*p* = 0.025), indicating that older patients tended to have lower educational-impact scores. No significant differences were found in the distribution of gender (*p* = 0.245) or the type of education received (VR vs. conventional, *p* = 0.29), suggesting these factors did not influence the likelihood of being in the limited-comprehension group vs. the higher-impact group within this sample. These results highlight the importance of age as a potential factor in the effectiveness of educational interventions and suggest that further research may be needed to tailor pacemaker education to older patients.

**Table 5 healthcare-12-00976-t005:** Comparison of Patient Knowledge Retention Between Conventional and VR Education Methods.

	Total		VR Enhanced		Conventional		*p*-Value
Total (N)	150		75		75		
Contact Info Retention (N; %)	73	48.7%	39	52.0%	34	45.3%	0.414
Anesthesia Type Knowledge (N; %)	142	94.7%	71	94.7%	71	94.7%	1
Pacemaker Location Awareness (N; %)	137	91.3%	70	93.3%	67	89.3%	0.384
Complication Awareness (N; %)	137	91.3%	70	93.3%	67	89.3%	0.563

Note: The table showcases the retention of specific knowledge elements by patients who received either conventional education or VR education, in preparation for a cardiac procedure. The total number (N) of patients and the percentage (%) correctly answering questions related to contact information, anesthesia type, pacemaker location, and common complications are reported, with a comparison between the conventional-education group and the VR-education group. The Chi-squared test was used to determine the *p*-values for differences in correct responses between groups.

## Data Availability

Data is available on request.

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
