# Peer review of "Comparing Conventional Physician-Led Education with VR Education for Pacemaker Implantation: A Randomized Study"

_healthcare, 2024, doi:10.3390/healthcare12100976_

Round 1

Reviewer 1 Report

Comments and Suggestions for Authors

The study examines an important and current topic, which can also be well utilized in patient care practice.

The abstract is informative and easy to follow.

The introduction clearly outlines the theoretical framework of the topic, the referenced studies are relevant, perhaps the presentation of the results related to VR education and its effectiveness in terms of various variables (gender, age, education level of patients) could be slightly expanded. The aim of the study is clear and sufficiently specific.

The methodological presentation of the research is thorough and professional, but at the same time to the point.

The presentation of the results is clear and easy to follow.

In the discussion, I propose a more detailed explanation of the age effect. In other studies, to what extent did older age prove to be an influencing factor with regard to VR education?

Conclusions are well supported by the results and discussion.

Overall, the study is a high-quality work that successfully supports the raison d'être of VR education.

Reviewer 2 Report

Comments and Suggestions for Authors

 Adela Drozdova et al. have conducted an original study ”Comparing Conventional Physician-Led Education with VR Education for Pacemaker Implantation: A Randomized Study”. The aim to prove feasibility and non-inferiority of patients’ education with the use of VR prior to the implantation of a  permanent pacemaker  and its beneficial effects on anxiety. The study presented a comparative evaluation of patient education methods for pacemaker implantation, focusing on traditional education versus VR immersion. Here are key aspects analyzed critically:

1.    The creation of a 360° VR video to simulate the hospital environment and procedure is innovative and potentially beneficial for patient education. The use of Oculus Meta Quest 2 headsets and ensuring content quality in the Czech language are commendable for enhancing patient comprehension and engagement.
2.    Line 119-121 ”Both groups of patients had a possibility to ask questions after receiving the education. They subsequently filled in a questionnaire, with questions marking their pre-hospital education, their understanding after our education and their experience with the VR” What are the questions? Were they open or closed? Why did you not use Cronbach's alpha as a measure of internal consistency?
3.    The study employed a structured questionnaire with a five-point scale to assess education quality and subjective VR experience. However, the evaluation focused primarily on immediate perceptions and understanding post-education, lacking longitudinal follow-up to determine lasting impact on patient outcomes or procedural complications.
4.    Notably, the study did not report adverse effects, but the short-term nature of the assessment may not capture long-term patient outcomes or procedural complications. Future research should incorporate follow-up assessments to determine the durability and real-world impact of VR-based education.
5.    This study demonstrates the potential of VR technology to enhance patient education and understanding in a procedural context. However, further research is needed to evaluate the sustained benefits of VR-based education, including its impact on patient outcomes and healthcare resource utilization. Cronbach's alpha is necessary to validate this study.

In summary, while the study highlights promising applications of VR in healthcare education, more comprehensive investigations are required to fully assess its efficacy, long-term benefits, and integration into routine clinical practice.
I recommend publishing the study after adding the questions and validating them.

Reviewer 3 Report

Comments and Suggestions for Authors

Dear Author/s,

The manuscript robustly outlines the study's context, design, and provides a clear insight into the initial findings. However, to further strengthen the manuscript and maximize its impact, I would like to suggest several areas for enhancement, in line with your original structured improvements.

The introduction clearly identifies specific problems—such as high anxiety levels among patients undergoing cardiac procedures and the consequences of this anxiety on health outcomes.

The objectives of the study are clear and relevant—demonstrating the feasibility and non-inferiority of VR education in reducing patient anxiety and improving understanding prior to pacemaker implantation.

The study's description as a prospective, randomized control trial with clear inclusion and exclusion criteria provides a good framework for research. Registration with ClinicalTrials.gov enhances the study's credibility and transparency.

Using SPSS for analysis and specifying non-parametric tests like the Mann-Whitney U test for continuous variables and Chi-square and Fisher’s exact test for categorical variables shows a methodologically sound approach.

Despite the detailed presentation, the results show no significant differences between the VR and conventional groups in many educational outcomes. This might suggest that VR, while innovative, does not significantly enhance understanding compared to conventional methods in this setting.

The results do not extensively discuss potential confounding factors that might influence the educational outcomes, such as prior familiarity with technology, which could affect how patients respond to VR education.

The discussion effectively connects the study's findings to broader trends in patient education, highlighting the general dissatisfaction with traditional educational methods and the potential of VR to improve engagement and information retention.

The call for further studies to confirm the non-inferiority of VR education in larger and potentially more diverse groups indicates a forward-looking approach, which is essential for the evolution of educational practices in healthcare.

The conclusions state the main outcomes of the study—namely, that VR education is non-inferior to traditional physician-led education in this patient population. Important nuances and context from the results lost due to the summary’s conciseness.

The unique features of VR that could contribute to its effectiveness, such as its interactive and immersive nature, are not elaborated upon. Discussing these features could help clarify why VR might result in better educational outcomes.

Recommendations for Improvement

1.             It broadly mentions VR's benefits but does not deeply connect these to the unique needs of patients undergoing this particular procedure.

2.             There is minimal insight into how the VR and conventional education methods will be compared, which is crucial for setting expectations about the study's design and integrity.

3.             The conclusion could emphasize patient engagement and feedback more in the development of the VR modules. Involving patients in the design process could ensure that the VR experience is tailored to their needs and concerns, making the study outcomes more robust and applicable. Consider highlighting patient involvement in the development of the VR tool to enhance the patient-centred approach of the study.

4.             While the study addresses potential VR-related adverse effects like dizziness and advises precautions, the actual impact of these side effects on the study's outcomes or the participants' experience is not discussed in depth.

5.             Considering adaptations of the VR content for other languages or cultural contexts in future studies could increase the generalizability of the findings.

6.             No adverse effects were reported related to VR, which might seem unlikely given the common issues with VR such as disorientation - especially in older people. We have no information on how long the instructional video lasted. This aspect deserves more thorough examination, especially given the known potential for side effects like VR-induced motion sickness. This could either indicate very effective mitigating measures or underreporting.

7.             While the discussion addresses the study's findings, it lacks a detailed comparison with existing literature, particularly how VR education's effectiveness in this setting compares with other studies involving younger cohorts or different medical conditions. Incorporate more comparisons with existing studies on VR in different patient populations or educational settings to contextualize your findings better within the broader research landscape.

8.             The discussion somewhat idealizes VR technology without sufficiently acknowledging its limitations or the barriers to its implementation, such as the need for technological infrastructure and patient familiarity with tech-based solutions. Although it mentions the need to emphasize certain information within the VR content, the discussion does not explore how the educational content can be optimized based on the study's findings to enhance understanding and retention further. Include a more balanced view of VR's limitations and the logistical or technical challenges associated with its implementation in clinical settings. While stating that VR is non-inferior, the section could expand on what this means in practical terms. For instance, discussing whether non-inferiority implies that VR should replace traditional methods or complement them. Elaborate on how VR can be integrated into clinical practice, considering both the benefits and the limitations observed. Discuss how VR might complement existing educational methods rather than replace them.

9.             Based on the findings, offer clear, actionable recommendations for healthcare providers on using VR in patient education, especially for older adults.

Regards

Reviewer

Comments on the Quality of English Language

While the content of the manuscript is strong, there are several areas where the quality of English could be improved to ensure that your findings are communicated as clearly and effectively as possible. Here are some specific suggestions: 1) I recommend a thorough review of subject-verb agreements, proper use of articles (a, an, the), and sentence structure; 2) attention to punctuation details, such as the use of commas, semicolons, and periods, can significantly enhance readability. 

Improving these aspects of the manuscript will not only enhance its readability but also its professional quality.

Round 2

Reviewer 2 Report

Comments and Suggestions for Authors

the authors have taken the interest to make the required changes and to correctly substantiate the study. The article can be published in this form